# Prioritized Task-Scheduling Algorithm in Cloud Computing Using Cat Swarm Optimization

**DOI:** 10.3390/s23136155

**Published:** 2023-07-05

**Authors:** Sudheer Mangalampalli, Sangram Keshari Swain, Tulika Chakrabarti, Prasun Chakrabarti, Ganesh Reddy Karri, Martin Margala, Bhuvan Unhelkar, Sivaneasan Bala Krishnan

**Affiliations:** 1School of Computer Science and Engineering, VIT-AP University, Amarvati 522237, Andhra Pradesh, India; ganesh.reddy@vitap.ac.in; 2School of Engineering and Technology, Centurion University of Technology and Management, Bhubaneswar 752050, Odisha, India; 3Department of Basic Sciences, Sir Padampat Singhania University, Udaipur 313601, Rajasthan, India; tulika.chakrabarti@spsu.ac.in; 4Department of Computer Science and Engineering, ITM SLS Baroda University, Vadodara 391510, Gujarat, India; drprasun.cse@gmail.com; 5School of Computing and Informatics, University of Louisiana at Lafayette, Lafayette, LA 70504, USA; martin.margala@louisiana.edu; 6Muma College of Business, University of South Florida Sarasota-Manatee campus, Sarasota, FL 33620, USA; bunhelkar@usf.edu; 7Singapore Institute of Technology, Singapore 139660, Singapore; sivaneasan@singaporetech.edu.sg

**Keywords:** cloud computing, task scheduling, makespan, energy consumption, SLA violation, OpenStack

## Abstract

Effective scheduling algorithms are needed in the cloud paradigm to leverage services to customers seamlessly while minimizing the makespan, energy consumption and SLA violations. The ineffective scheduling of resources while not considering the suitability of tasks will affect the quality of service of the cloud provider, and much more energy will be consumed in the running of tasks by the inefficient provisioning of resources, thereby taking an enormous amount of time to process tasks, which affects the makespan. Minimizing SLA violations is an important aspect that needs to be addressed as it impacts the makespans, energy consumption, and also the quality of service in a cloud environment. Many existing studies have solved task-scheduling problems, and those algorithms gave near-optimal solutions from their perspective. In this manuscript, we developed a novel task-scheduling algorithm that considers the task priorities coming onto the cloud platform, calculates their task VM priorities, and feeds them to the scheduler. Then, the scheduler will choose appropriate tasks for the VMs based on the calculated priorities. To model this scheduling algorithm, we used the cat swarm optimization algorithm, which was inspired by the behavior of cats. It was implemented on the Cloudsim tool and OpenStack cloud platform. Extensive experimentation was carried out using real-time workloads. When compared to the baseline PSO, ACO and RATS-HM approaches and from the results, it is evident that our proposed approach outperforms all of the baseline algorithms in view of the above-mentioned parameters.

## 1. Introduction

Cloud computing is a distributed computing model that renders on-demand computing and storage services (among other services) to their customers based on their needs. According to NIST [1], cloud computing can be defined as, “on demand, network access to a shared pool of configurable computational resources”, which gives services to cloud users. This paradigm consists of different deployment models, i.e., public, private and hybrid clouds.

Figure 1 represents various deployment models in the cloud paradigm, where the public cloud model leverages services to all cloud users around the globe. The private cloud model leverages services to users where its application resides in a particular organization, and the hybrid cloud model provides services to users, in which some of the services are provided publicly and some services are provided privately. To effectively provision resources to users, the cloud provider needs to employ an effective task scheduler for seamless provisioning and deprovisioning of resources. Users of cloud computing are vast and diversified, and it is a challenging task to map the diversified and heterogeneous requests from various users onto virtual resources. An ineffective task scheduler will reduce the quality of service of the cloud service, and increase the makespan and energy consumption, which also leads to SLA violation, affecting both cloud providers and consumers. Many authors have solved task scheduling problems in cloud computing using metaheuristic algorithms, e.g., PSO [2], GA [3], and ACO [4]. All these are metaheuristic approaches, and among these approaches, some of them work based on swarm updating, pheromone updating, and chromosome updating techniques. Previous authors have used these mechanisms to solve task scheduling in this paradigm, but there is still a chance to improve the scheduling pattern in this paradigm because it is an NP-hard problem. Therefore, we can still improve the effectiveness of the scheduler by taking the priorities of the tasks dispersed on the cloud interface and calculating the priorities for the VM based on electricity price unit cost. Based on these priorities, the scheduler needs to take decisions by the mapping of tasks onto appropriate VMs. In this paper, we used cat swarm optimization [5] to tackle task scheduling in the cloud paradigm. 

### Motivation and Contributions

The main motivation to carry out this research work is to effectively schedule virtual resources for various heterogeneous cloud users with a good quality of service while minimizing energy consumption in datacenters and SLA violations between cloud users and the provider. Scheduling is a highly challenging scenario in the cloud paradigm as there are a variable number of customers requesting resources, and the cloud provider needs to provide services by employing an effective scheduling algorithm according to their needs. However, in real time it is a huge challenge for a cloud provider to provision resources based on the types of task that require cloud services. Therefore, in our research we have carefully identified the suitability of tasks by calculating priorities and then fed those priorities to the scheduler, generating scheduling decisions accordingly.

The contributions of this paper are presented below:A prioritized task-scheduling algorithm is developed using cat swarm optimization [5];The assignment of tasks to VMs in a scheduling model by calculating the priorities of the tasks;A synthetic workload is given as input to the algorithm to conduct simulations;SLA violation, makespans, and energy consumption parameters are addressed in this approach using real-time workloads.

The remaining manuscript is organized as follows: Literature Survey is represented in Section 2, the problem statement and Proposed System Architecture are represented in Section 3, Proposed Methodology is represented in Section 4, Simulations and Results are presented in Section 5, and Conclusion and Future Work is presented in Section 6.

## 2. Literature Survey

In [6], the authors proposed a task-scheduling approach that addresses parameters, i.e., resource utilization, energy, SLA violation. It was modelled by using the CSSA mechanism. It was evaluated using GA-PSO, SSA, PSO-BAT approaches. The results have shown that the abovementioned parameters were greatly minimized for proposed approach. In [7], the CSA algorithm proposed by the authors maps tasks to the VM by minimizing the makespan. Crow Search algorithm used for solving scheduling. It was evaluated against the existing Min-Min and ACO algorithms. The proposed CSA outperforms existing approaches for the specified metrics for diversified workloads. 

The authors in [8] developed a resource allocation mechanism intended to allow vehicular cloud architecture to offload requests while on boarding vehicles and avoiding latency for processing of requests. HAPSO was used as methodology for solving resource allocation in cloud paradigm. Vehicular network implementation using SUMO simulator and cloud simulation was achieved on Matlab. It was compared against existing PSO, self-adaptive PSO and HAPSO, showing a significant reduction in the makespan and energy consumption. In [9], the authors proposed a hybridized approach, LJFP-MCT combined with PSO to schedule tasks to appropriate VMs. It was compared to PSO, variations of PSO and MCT approaches. LJFP-MCT outperforms existing algorithms for the minimization of makespans and degrees of imbalance. 

HIGA is a hybridized task-scheduling algorithm proposed by the authors in [10], which addresses makespan, energy consumption and execution overhead in cloud datacenters. The methodology used in this approach is a combination of harmony-inspired and GA algorithms. It was compared to various existing approaches. From the results, it dominated benchmark algorithms for specified parameters. An energy-based task-scheduling algorithm was proposed by the authors in [11] for the minimization of makespans and energy consumption in cloud datacenters. BWF and TOPSIS algorithms were hybridized to address scheduling problem in cloud computing. Initially TOPSIS was used to identify prioritized group of tasks for its execution, and later, BWF used as scheduling criteria. It was evaluated against BWF, TOPSIS and PSO approaches. Simulation results showed that it performed better than existing mechanisms for different parameters. 

The authors of [12] proposed a scheduling algorithm, which addresses makespans and energy consumption. The methodology used in this approach is combination of GA and BFA. It was assessed in comparison to GA, PSO and BFA. From a simulation, it was shown to have a greater impact compared to existing mechanisms for the abovementioned parameters. A task-scheduling algorithm using the inverted ACO mechanism was proposed by [13]. Simulations were conducted on Cloudsim. It was evaluated against different PSO variations. Inverted ACO dominates existing algorithms in terms of energy consumption, response time and SLA violations.

In [14], a task-scheduling mechanism was proposed that uses a combination of MVO and PSO algorithms. The aim of this approach is to address makespans and the utilization of resources. It showed a greater impact compared to the baseline mechanism for specified metrics. A task-scheduling and load-balancing algorithm was proposed in [15], which focuses on makespans and load balance during task distribution. CSSA methodology was used to tackle task scheduling. It was evaluated against PSO and ABC approaches. From the results, it outperforms existing algorithms in the minimization of makespans and load balance during task distribution. 

PCGWO, a task-scheduling algorithm proposed to tackle makespans, cost, and deadlines, was proposed in [16]. It was modelled based on improvement made to the GWO algorithm. It was assessed in relation to existing FCFS and GWO approaches. The results shows a greater impact than baseline mechanisms for specified parameters. A hybridized approach, i.e., MSDE proposed in [17], was intended to minimize makespans. The methodology used in this approach was a combination of a Moth search with a DE parameter. It was implemented using Matlab tool 2022a. Random and synthetic workloads were given as the input to the proposed approach to evaluate the parameter, i.e., makespans. It was compared baseline mechanisms, with the results showing a superior impact for specified parameters. The MVO-GA task-scheduling mechanism is a hybrid approach proposed in [18]. It is a combination of MVO and GA algorithms. The parameters addressed by the proposed approach are service availability and scalability. It was implemented using MATLAB tool by simulating a cloud environment. It was evaluated against the baseline approaches, i.e., MVO, GA and PSO. From the simulation, MVO-GA showed its dominance over the baseline algorithms. In [19], a hybrid task-scheduling framework was proposed based on ACO-Fuzzy approaches. It was used to effectively distribute, compute and network resources to end users. ACO was used to explore the local search mechanism based on pheromone updating, while fuzzy controller makes a scheduling decision based on the workload approach [20]. It was assessed by comparing it to existing ACO and PSO scheduling approaches. The results showed that the ACO-Fuzzy mechanism [21,22,23] outperforms existing algorithms, minimizing end user costs. SLA violation and power consumption are to be considered as important parameters in cloud paradigms and need to be optimized by using an effective task-scheduling model. The authors of [24] addressed the abovementioned parameters by using the crowding entropy mechanism, which hybridizes it with PSO. It was implemented on MATLAB and compared to GA and ACO algorithms. The results revealed that VMPMOPSO showed dominance over existing the mechanisms. In [25], SLNO was proposed by authors as a task-scheduling mechanism consisting of both exploration and exploitation capabilities. It aims at minimize task completion, energy consumption and overall cost. Sea lion optimization methodology was used to model the scheduling mechanism. It was assessed in relation to WOA, GWO and RR mechanisms using an extensive set of workloads. The results proved that SLNO outperformed the existing algorithms. The authors of [26] proposed a multi objective scheduling model focused on makespans and degrees of imbalance. VWOA was evaluated against [27] WOA, RR approaches and it dominated the abovementioned approaches for said parameters. In [28], the authors proposed a distributed optimization scheduler for heterogeneous cloud resources using different functions, i.e., linear, sigmoid and deadline. This approach was implemented on a test bed running on Google cluster with a deep reinforcement learning approach and was finally compared to existing baseline approaches. The proposed DO4A outperforms existing algorithms in the minimization of job processing capacity and transmission delay. In [29], the authors proposed a microservice resource allocation framework that adapts to the respective workflows to optimize response time. This approach uses a reinforcement learning approach to identify the type of workflow, and based on that, it will manage resources effectively, minimizing response time.

Table 1 shows many of the existing task scheduling algorithms that use various nature inspired algorithms and many of the authors used parameters such as makespan, execution time, energy consumption, and SLA violations but failed due to addressing parameter combinations of makespan, energy consumption and SLA violations as ineffective at provisioning resources to users, as a scheduler affects makespan and energy consumption directly, and SLA violations indirectly. Therefore, there is a relationship between makespan, energy consumption and SLA violation. Our proposed approach addresses all these metrics while considering the priorities of tasks, VMs and schedule resources accordingly.

## 3. Proposed System Architecture

This section precisely discusses the proposed system architecture in a detailed manner. Assume we took n tasks, indicated as $t_{n}=\{t_{1},t_{2},\ldots .t_{n}\}$, k VMs indicated as $v_{k}=\{v_{1},v_{2},v_{3}\ldots \ldots \ldots v_{k}\}$, j hosts indicated as $h_{j}=\{h_{1},h_{2},\ldots .h_{j}\}$, i datacenters, i.e., $d_{i}=\{d_{1},d_{2},d_{3}\ldots .d_{i}\}$. The problem is defined here as n tasks are carefully mapped on to k VMs residing in j hosts and in i datacenters while minimizing SLA violations, energy consumption and makespans. Table 2 below indicates notations used in the proposed system architecture for mathematical modeling.

Figure 2 shows the proposed system architecture. In Figure 2, various cloud users first submit requests to the cloud console. The cloud broker will take those requests and submit them to the task manager. The task manager has to check whether the requests made by the users are valid or not based on SLA. After verifying the users’ requests, the task manager feeds all requests to the scheduler in the generalized architecture. In the proposed system architecture, after the users’ request submissions from cloud users are escalated to the task manager level, priorities of tasks calculated initially based on length, runtime processing capacities of tasks. After calculating the tasks, VM priorities are calculated based on the electricity cost at the datacenter’s location. Upon capturing of these priorities, ranking are given for all tasks and fed to the scheduler to assign tasks effectively on suitable VMs. Therefore, in order to map tasks appropriately on to VMs, we need to minimize makespans, energy consumption and SLA violations.

To calculate task priority, we initially calculate the current load of the VMs. The overall load of the VMs is calculated using Equation (1).
(1)$$l^{vm}=\sum l^{k}$$
where $l^{vm}$ indicates current load of k VMs.

After calculating the load of the VMs, we evaluate the load on the hosts, which is calculated using Equation (2).
(2)$$l^{h}=l^{vm}/\sum h_{j}$$
where $l^{h}$ indicates overall load on physical hosts.

After calculating the loads of the VMs and physical hosts but before defining priority of tasks, we need to check the processing capacity of the VMs as it is very important in our scheduling criteria to map suitable tasks to the appropriate VMs. Therefore, the VM processing capacity is calculated using Equation (3).
(3)$$pr^{vm}=pr^{no}\times pr^{mips}$$
where $pr^{vm}$ indicates the VM processing capacity, $pr^{no}$ indicates the number of processing elements, and $pr^{mips}$ indicates the computational speed of a VM.

The VM processing capacity is calculated by using Equation (4).
(4)$$tot_{pr}^{vm}=\sum pr^{vm}$$

After calculating the VM processing capacity, we now need to calculate size of task, which is evaluated using Equation (5).
(5)$$t_{n}^{size}=t^{mips}\times t^{p}$$

Now, we can calculate the priority of tasks using Equation (6) below.
(6)$$t^{pr}=t_{n}^{size}/pr^{vm}$$

In our research, we are not only calculating the priority of tasks, but we are also identifying the priorities of the VMs based on the unit electricity cost at datacenter’s location. The higher unit electricity cost of a datacenter gives less priority to schedule tasks onto high-prioritized VMs, which has lower electricity unit cost through which we minimize makespans, energy consumption and SLA violations.
(7)$$vm^{pr}=\frac{high^{unit\;elect\;cost}}{d_{i}^{unit\;elect\;cost}}$$
where $high^{unit\;elect\;cost}$ indicates the highest unit cost of electricity price considered in all datacenters and $d_{i}^{unit\;elect\;cost}$ indicates the unit cost of electricity price at a particular datacenter.

After evaluating both the task and VM priorities, our main research objective is now minimizing makespans, SLA violations and energy consumption.

Makespan is the execution time of a task when run on a VM. It is calculated using Equation (8) below.
(8)$$ms^{n}=avail^{k}+e^{n}$$
where $ms^{n}$ indicates the makespan of n tasks, $e^{n}$ indicates the execution time of n tasks and $avail^{k}$ indicates the availability of k VMs.

Our next parameter to model for this scheduler is energy consumption, which is an important parameter from the perspectives of both the cloud provider and consumer. Energy consumption in cloud paradigms consists of two parts: one part indicates the consumption of energy during computation and the other part indicates the consumption of energy when idling. It is identified using Equation (9) below.
(9)$$e_{con}\left(vm^{k}\right)=\int_{0}^{k}e_{con}^{com}\left(vm^{k}t\right)+e_{con}^{idle}\left(vm^{k}t\right)dt$$

After calculating the energy consumption of the VMs during computation and when idling, we can now calculate the overall energy consumption of all VMs, which is calculated using Equation (10) below.
(10)$$e_{con}=\sum e_{con}(vm^{k})$$

After calculating the makespan and energy consumption, we have to calculate SLA violations, which is an important metric for both the cloud consumer and provider because if SLA is violated at a particular instance of time by not completing a task with in its deadline, it will lead to performance degradation. Now, to calculate SLA violations, we first calculate the active time of the physical host and performance degradation. It is calculated using Equations (11) and (12), respectively.
(11)$$ACT=\frac{1}{r}\sum_{m=1}^{r}\begin{array}{c} \frac{violation\;tim{e_{h}}_{j}}{AC{T_{h}}_{j}} \end{array}$$
(12)$$Per_{dg}=\frac{1}{k}\sum_{b=1}^{k}\frac{Per_{dg}^{r}}{tot_{r}^{vm}}$$

From Equations (11) and (12) above, we have calculated the active time of the physical hosts and performance degradation. From both Equations (11) and (12), we can calculate SLA violations using Equation (13) below.
(13)$$sla_{violation}=ACT*Per_{dg}$$

Now, we have identified the metrics and calculated them using Equations (8), (10) and (13). We now need to define a fitness function to optimize our parameters using cat swarm optimization. Fitness function calculated using below Equation (14).
(14)$$f\left(x\right)=min\sum ms^{n}\left(x\right),e_{con}\left(x\right),sla_{violation}(x)$$

In Section 3, we clearly presented the mathematical modeling and proposed system architecture, and in next section, we present the methodology used to model our proposed prioritized scheduler in a detailed manner.

## 4. Methodology and Proposed Prioritized Task Scheduler Using Cat Swarm Optimization

### 4.1. Cat Swarm Optimization

This section presents a brief overview of the cat swarm optimization algorithm presented in [5]. It’s nature inspired the algorithm used as the methodology in our research. This algorithm works based on the behavior of cats in nature. Cats have two modes: seeking and active. The seeking mode refers to when a cat is at rest but is still ready and alert for any kind of task given to that cat, whereas active mode refers to the chasing of prey. In this algorithm, cats in active mode chase for a particular prey for certain amount of time. This process continuously happens until iterations are completed. For this to happen, cats are first initialized randomly by evolving swarm, and before that, all cats are divided into two groups, i.e., they are separated by seeking and active modes. For every cat, which is in active mode, a fitness value needs to be calculated for every iteration. After the initialization of the cats, the velocity for all cats are calculated using Equation (15) below.
(15)$$ve_{d}^{q}\left(t+1\right)=s*ve_{d}^{q}\left(t\right)+b*u*(x_{best}^{d}-x_{q}^{d})$$
where $ve_{d}^{q}\left(t\right)$ is the velocity of the qth cat at tth iteration, $x_{best}^{d}$ is best solution for that iteration, *u* is a random number that lies in 0 and 1, and *b* is a constant.

Updating of the cat’s position in the solution space is calculated using Equation (16).
(16)$$x_{q}^{d}\left(t+1\right)=x_{q}^{d}+ve_{d}^{q}(t+1)$$

The calculation of velocity and updating of the cat’s positions are to be calculated until all iterations have been completed.

### 4.2. Proposed Prioritized Task Scheduling Algorithm Using Cat Swarm Optimization

The below section presents the proposed task scheduling approach in Algorithm 1.

**Algorithm 1** Prioritized Task Scheduling Algorithm Using Cat Swarm Optimization**Input:**   $t_{n}=\{t_{1},t_{2},\ldots .t_{n}\},\;v_{k}=\{v_{1},v_{2},v_{3}\ldots \ldots \ldots v_{k}\},\;h_{j}=\{h_{1},h_{2},\ldots .h_{j}\},\;d_{i}=\{d_{1},d_{2},d_{3}\ldots .d_{i}\}.$**Output:** Generation of schedules by considering priorities with optimization of $ms^{n},\;e_{con}\;\mathrm{and}\;sla_{violation}$

StartInitialize $\;t_{n}=\{t_{1},t_{2},\ldots .t_{n}\},\;v_{k}=\{v_{1},v_{2},v_{3}\ldots \ldots \ldots v_{k}\},\;h_{j}=\{h_{1},h_{2},\ldots .h_{j}\},\;d_{i}=\{d_{1},d_{2},d_{3}\ldots .d_{i}\}.$ // tasks, VMs, physical hosts, data centers values initialized //Initialize cat opulation generationFor each $t_{n},v_{k}$Calculate incoming task priorities using Equation (6).Calculate VM priorities using Equation (7).Calculate fitness function by Equation (14).Calculate velocity of cats population using Equation (15)Update its global fitness value.Calculate parameters using Equations (8), (10) and (13).Check best fitness value appeared or not using Equation (15)Check parameter values for minimizationOtherwise update cats position using Equation (16) and continue the process from Equation (4)Repeat this process till all iterations completedStop


### 4.3. Time Complexity of Prioritized CSO

For Algorithm 1 above, we initially generated a random cat population and needed to generate N tasks and D VMs as resources, so time complexity is O(*N*D*). The time complexity of calculating VM priorities is O(*n*), where n is number of VMs. The time complexity for calculating the fitness function is O(*m*), where m is the complexity of fitness function. The time complexity of updating and minimizing values in the fitness function is O(*1*). For one iteration, time complexity is O(*N*D*) + O(*n + m + 1*), but O(*N*D*) is much larger than O(*n + m + 1*). We can approximate it to O(*N*D*). Therefore, the total time complexity for T iterations is O(*N*D*T*).

## 5. Simulations and Results

This section presents the overall simulation and results in a detailed manner. The entire simulation was carried out on a discrete event simulator named Cloudsim, which creates a cloud environment based on the Java programming language. For efficient evaluation of the parameters, we have given HPC2N [21] and NASA [22] parallel work logs as input to our algorithm. After evaluating our proposed prioritized CSO in a simulated environment, we created a real-time test bed in an OpenStack cloud environment to check the efficacy of our approach. Initially we used nova compute service to launch our VM. VM initialization was executed using Glance service, so we used a basic Linux VM, to which we gave a random generated workload and the input from both the HPC2N and NASA workloads, then identified the efficacy for the abovementioned parameters.

### 5.1. Simulation Settings

This entire simulation runs on a system with a configuration comprising an i5 processor, 32 GB RAM and 1024 GB hard disk capacity. We used a Linux operating system to run this simulation and installed the Cloudsim tool. Below, Table 3 represents settings used in our simulation.

### 5.2. Makespan Evaluation

Initially, as per our discussion in mathematical modeling, we calculated makespan in this research. It was evaluated against HPC2N and NASA workloads and compared to baseline algorithms, such as PSO and ACO. From the results, our proposed prioritized cat scheduler shows significant impact on SOTA approaches by minimizing the makespan.

Table 4 below shows the makespan calculation for PSO, ACO, RATS-HM and prioritized CSO for 100, 500 and 1000 tasks using the HPC2N workload. The makespans generated for PSO for various 100, 500 and 1000 tasks are 1358.9, 1756.9 and 2067.2, respectively. The makespans generated for ACO for various 100, 500 and 1000 tasks are 1364.8, 1784.9 and 2245.9, respectively. The makespans generated for RATS-HM for various 100, 500 and 1000 tasks are 1486.32, 1856.18 and 2563.9, respectively. The makespans generated for prioritized CSO for various 100, 500 and 1000 tasks are 1276.9, 1356.5 and 1856.8, respectively. From results displayed in Table 4 and Figure 3 below, it is evident that the prioritized CSO scheduler better minimized makespans when compared to PSO, ACO and RATS-HM

Table 5 below shows the makespan calculation for PSO, ACO, RATS-HM and prioritized CSO for 100, 500 and 1000 tasks using the HPC2N workload in an OpenStack cloud. The makespans generated for PSO for various 100, 500 and 1000 tasks are 1467.7, 1768.5 and 2156.8, respectively. The makespans generated for ACO for various 100, 500 and 1000 tasks are 1387.23, 1894.36 and 2256.72, respectively. The makespans generated for RATS-HM for various 100, 500 and 1000 tasks are 1567.12, 1923.98 and 2734.26, respectively. The makespans generated for prioritized CSO for various 100, 500 and 1000 tasks are 1345.35, 1467.12 and 1756.21, respectively. From results displayed in Table 5 and Figure 4 below, it is evident that the prioritized CSO scheduler better minimized makespans when compared to PSO, ACO and RATS-HM.

Table 6 below shows the makespan calculation for PSO, ACO and prioritized CSO for various 100, 500 and 1000 tasks using the NASA workload. The makespans generated for PSO for various 100, 500 and 1000 tasks are 659.2, 1287.5 and 1356.8, respectively. The makespans generated for ACO for various 100, 500 and 1000 tasks are 785.6, 856.9 and 1187.92, respectively. The makespans generated for RATS-HM for various 100, 500 and 1000 tasks are 843.98, 756.18 and 1098.2, respectively. The makespans generated for prioritized CSO for various 100, 500 and 1000 tasks are 523.67, 659.45 and 878.23, respectively. From the results displayed in Table 6 and Figure 5 below, it is evident that the prioritized CSO scheduler better minimized the makespan when compared to PSO, ACO and RATS-HM.

Table 7 below shows the makespan calculation for PSO, ACO and prioritized CSO for various 100, 500 and 1000 tasks using the NASA workload in an OpenStack cloud. The makespans generated for PSO for various 100, 500 and 1000 tasks are 876.32, 1478.12 and 1875.11, respectively. The makespans generated for ACO for various 100, 500 and 1000 tasks are 923.45, 1075.32 and 1256.8, respectively. The makespans generated for RATS-HM for various 100, 500 and 1000 tasks are 1078.57, 1245.32 and 1467.21, respectively. The makespans generated for prioritized CSO for various 100, 500 and 1000 tasks are 756.21, 619.17 and 945.67, respectively. From results displayed in Table 7 and Figure 6 below, it is evident that the prioritized CSO scheduler better minimized the makespan when compared to PSO, ACO, RATS-HM.

### 5.3. Energy Consumption Evaluation

After calculating makespan, we calculated energy consumption in this research. It was evaluated against HPC2N and NASA workloads, and compared to baseline algorithms, such as PSO and ACO. From the results, our proposed prioritized cat scheduler showed greater impact when compared to existing approaches regarding minimizing energy consumption. Table 8 below shows the energy consumption calculation for PSO, ACO and prioritized CSO for various 100, 500 and 1000 tasks using the HPC2N [22] workload. The energy consumptions generated for PSO for various 100, 500 and 1000 tasks are 47.87, 98.65 and 145.98, respectively. The energy consumptions generated for ACO for various 100, 500 and 1000 tasks are 38.98, 87.56 and 123.98, respectively. The energy consumptions generated for RATS-HM for various 100, 500 and 1000 tasks are 42.78, 67.36,133.97, respectively. The energy consumptions generated for prioritized CSO for various 100, 500 and 1000 tasks are 28.78, 43.99 and 108.99, respectively. From the results displayed in Table 8 and Figure 7 below, it is evident that the prioritized CSO scheduler better minimized energy consumption when compared to PSO, ACO and RATS-HM.

Table 9 below shows the energy consumption calculation for PSO, ACO and prioritized CSO for various 100, 500 and 1000 tasks using the HPC2N workload in an OpenStack cloud. The energy consumptions generated for PSO for various 100, 500 and 1000 tasks are 56.15, 104.32, 157.12, respectively. The energy consumptions generated for ACO for various 100, 500 and 1000 tasks are 42.15, 88.23 and 135.67, respectively. The energy consumptions generated for RATS-HM for various 100, 500 and 1000 tasks are 56.18, 72.18 and 142.78, respectively. The energy consumptions generated for prioritized CSO for various 100, 500 and 1000 tasks are 31.67, 45.19 and 98.45, respectively. From the results displayed in Table 9 and Figure 8 below, it is evident that the prioritized CSO scheduler better minimized energy consumption when compared to PSO, ACO and RATS-HM.

Table 10 below shows the energy consumption calculation for PSO, ACO and prioritized CSO for various 100, 500 and 1000 tasks using the NASA [23] workload. The energy consumptions generated for PSO for various 100, 500 and 1000 tasks are 49.56, 85.79, 112.79, respectively. The energy consumptions generated for ACO for various 100, 500 and 1000 tasks are 38.78, 61.56 and 124.89, respectively. The energy consumptions generated for RATS-HM for various 100, 500 and 1000 tasks are 56.12, 64.37 and 135.88, respectively. The energy consumption generated for prioritized CSO for various 100, 500 and 1000 tasks are 22.98, 32.32 and 99.56, respectively. From the results displayed in Table 10 and Figure 9 below, it is evident that the prioritized CSO scheduler better minimized energy consumption when compared to PSO, ACO and RATS-HM.

Table 11 below shows the energy consumption calculation for PSO, ACO and prioritized CSO for various 100, 500 and 1000 tasks using the NASA workload in an OpenStack cloud. The energy consumptions generated for PSO for various 100, 500 and 1000 tasks are 52.44, 89.67 and 118.43, respectively. The energy consumptions generated for ACO for various 100, 500 and 1000 tasks are 49.56, 72.19 and 132.18, respectively. The energy consumption generated for RATS-HM for various 100, 500 and 1000 tasks are 59.15, 71.25 and 156.28, respectively. The energy consumptions generated for prioritized CSO for various 100, 500 and 1000 tasks are 26.74, 30.16,87.34, respectively. From the results displayed in Table 11 and Figure 10 below, it is evident that the prioritized CSO scheduler minimized energy consumption when compared to PSO, ACO and RATS-HM.

### 5.4. SLA Violation Evaluation

After calculating makespan and energy consumption, we calculated SLA violations in this research. It was evaluated against HPC2N and NASA workloads and compared to baseline algorithms, such as PSO and ACO. From the results, our proposed prioritized cat scheduler shows greater impact when compared to existing approaches regarding minimizing SLA violations.

Table 12 below shows the SLA violation calculation for PSO, ACO and prioritized CSO for various 100, 500 and 1000 tasks using the HPC2N workload. The SLA violations generated for PSO for 100, 500 and 1000 tasks are 15, 21 and 31, respectively. The SLA violations generated for ACO for various 100, 500 and 1000 tasks are 17, 20 and 35, respectively. The SLA violations generated for RATS-HM for various 100, 500 and 1000 tasks are 18, 23 and 21, respectively. The SLA violation generated for prioritized CSO for various 100, 500 and 1000 tasks are 7, 11 and 12, respectively. From the results displayed in Table 12 and Figure 11 below, it is evident that the prioritized CSO scheduler better minimized SLA violations when compared to PSO, ACO and RATS-HM.

Table 13 below shows the SLA violation calculation for PSO, ACO and prioritized CSO for various 100, 500 and 1000 tasks using the HPC2N workload for an OpenStack cloud. The SLA violations generated for PSO for 100, 500 and 1000 tasks are 18, 27 and 38, respectively. The SLA violations generated for ACO for various 100, 500 and 1000 tasks are 21, 36 and 39, respectively. The SLA violations generated for RATS-HM for various 100, 500 and 1000 tasks are 31, 26 and 25, respectively. The SLA violation generated for prioritized CSO for various 100, 500 and 1000 tasks are 9, 14 and 11, respectively. From the results displayed in Table 13 and Figure 12 below, it is evident that prioritized CSO scheduler better minimized SLA violations when compared to PSO, ACO and RATS-HM.

Table 14 below shows the SLA violation calculation for PSO, ACO and prioritized CSO for various100, 500 and 1000 tasks using the NASA workload. The SLA violations generated for PSO for various 100, 500 and 1000 tasks are 11, 18 and 21, respectively. The SLA violations generated for ACO for various 100, 500 and 1000 tasks are 14, 10 and 19, respectively. The SLA violations generated for RATS-HM for various 100, 500 and 1000 tasks are 16, 12 and 21, respectively. The SLA violations generated for prioritized CSO for various 100, 500 and 1000 tasks are 4, 9 and 11, respectively. From the results displayed in Table 14 and Figure 13 below, it is evident that prioritized CSO scheduler better minimized SLA violations when compared to PSO, ACO and RATS-HM.

Table 15 below shows the SLA violation calculation for PSO, ACO and prioritized CSO for various 100, 500 and 1000 tasks using the NASA workload in an OpenStack cloud. The SLA violations generated for PSO for various 100, 500 and 1000 tasks are 15, 21 and 29, respectively. The SLA violations generated for ACO for various 100, 500 and 1000 tasks are 21, 18 and 16, respectively. The SLA violations generated for RATS-HM for various 100, 500 and 1000 tasks are 19,18 and 25, respectively. The SLA violations generated for prioritized CSO for various 100, 500 and 1000 tasks are 6, 10 and 14, respectively. From the results displayed in Table 15 and Figure 14 below, it is evident that the prioritized CSO scheduler better minimized SLA violations when compared to PSO, ACO and RATS-HM.

### 5.5. Discussion of Results of Simulation and in OpenStack Cloud Environment

After simulating and implementing the results in an OpenStack cloud environment with different approaches, we evaluated the results and calculated the improvement of the results compared to those of existing approaches. For experimentation purposes, we used standard worklogs captured from HPC2N and NASA, and these workloads were fed to our scheduler, which ran for 100 times. Detailed analysis of results and improvements in SLA violations, energy consumption, makespans are provided in Table 16, Table 17, Table 18, Table 19, Table 20 and Table 21 below.

## 6. Conclusion and Future Work

Cloud computing is a distributed paradigm that leverages on-demand services to users based on their application needs. For the effective provisioning of services to cloud users, cloud providers need to employ an effective task scheduling mechanism, which should map incoming tasks onto a cloud interface and to appropriate VMs in the cloud paradigm. In this manuscript, we propose an approach, which considers the priorities of tasks and priorities based on unit electricity cost at the datacenter locations. Existing authors used various metaheuristic algorithms to solve scheduling problems in cloud paradigms but these metaheuristic approaches only provide near-optimal solutions. Still, there is a chance to improve scheduling process by evaluating priorities and feeding the workload to the scheduler to generate scheduling decisions. We used cat swarm optimization to solve task scheduling problems in this paradigm. Extensive simulations are carried out on Cloudsim. Simulations were conducted by using HPC2N and NASA parallel work logs. They were evaluated against existing PSO and ACO approaches. From the simulation results, it has been proved that the proposed approach outperforms existing algorithms by minimizing makespans, energy consumption, SLA violations. In the future, we will employ a machine learning framework to predict the type of workloads coming onto cloud interface to provide and generate effective schedules to various heterogeneous users.

## Figures and Tables

**Figure 1 sensors-23-06155-f001:**
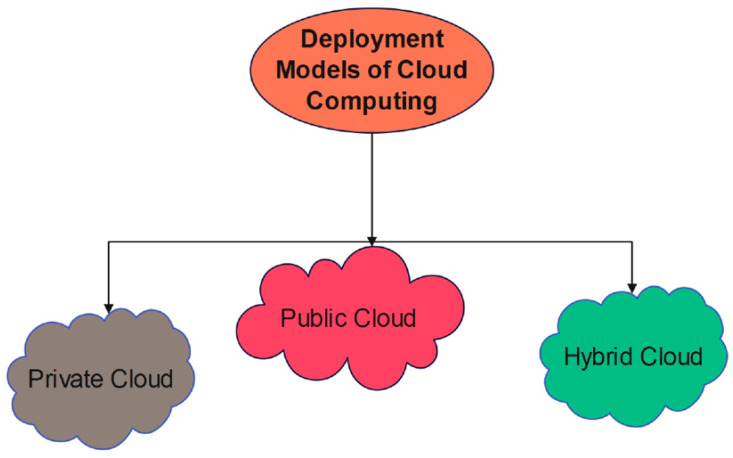
Deployment models of cloud computing.

**Figure 2 sensors-23-06155-f002:**
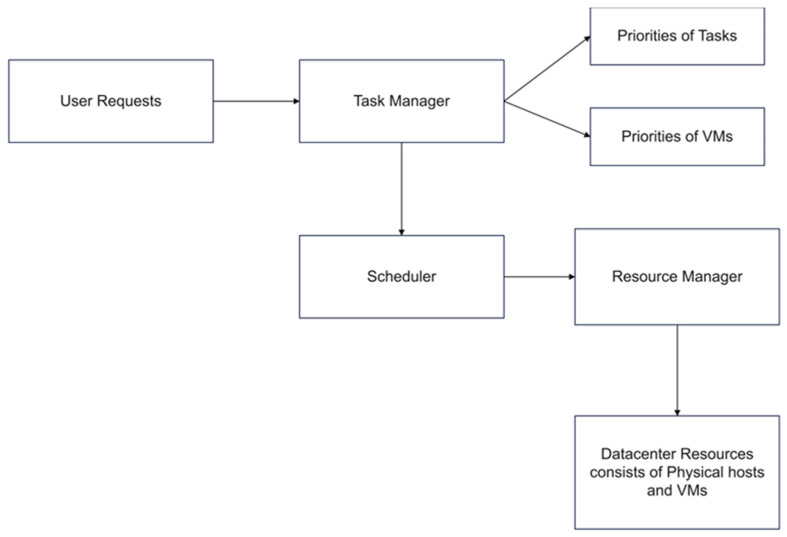
Proposed system architecture.

**Figure 3 sensors-23-06155-f003:**
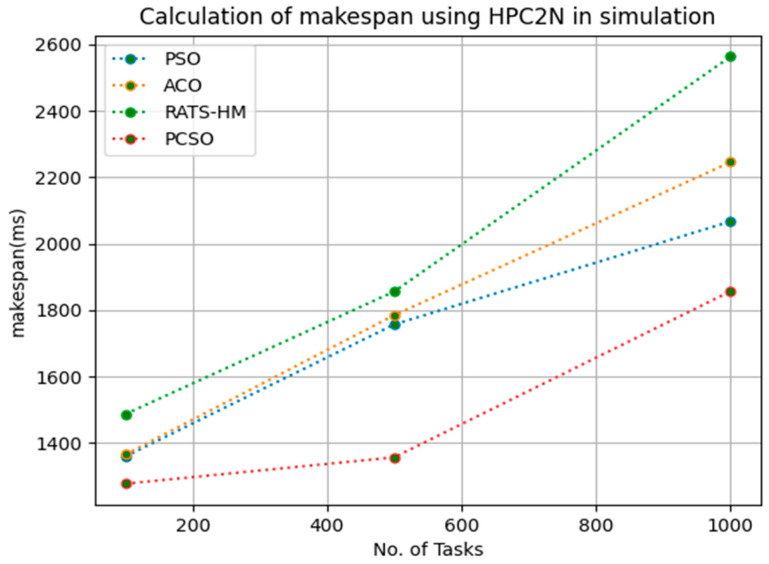
Evaluation of makespan HPC2N in Simulation.

**Figure 4 sensors-23-06155-f004:**
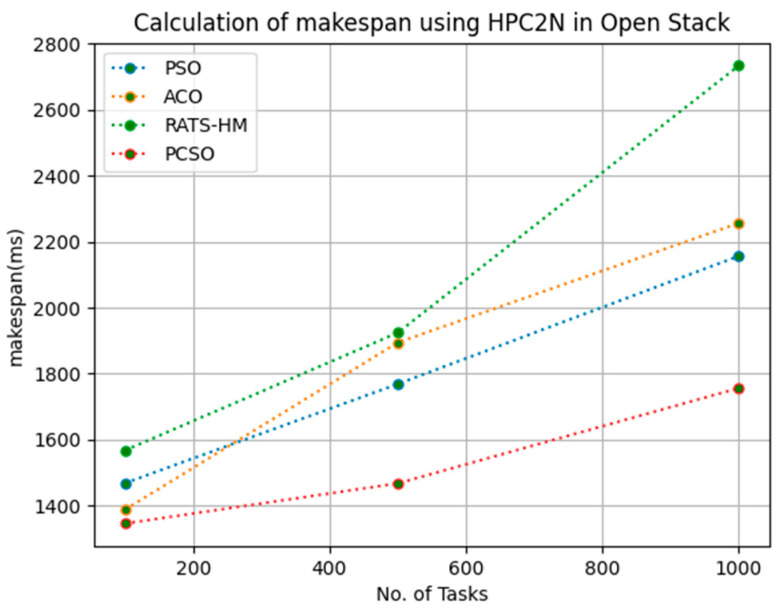
Evaluation of makespan HPC2N in OpenStack.

**Figure 5 sensors-23-06155-f005:**
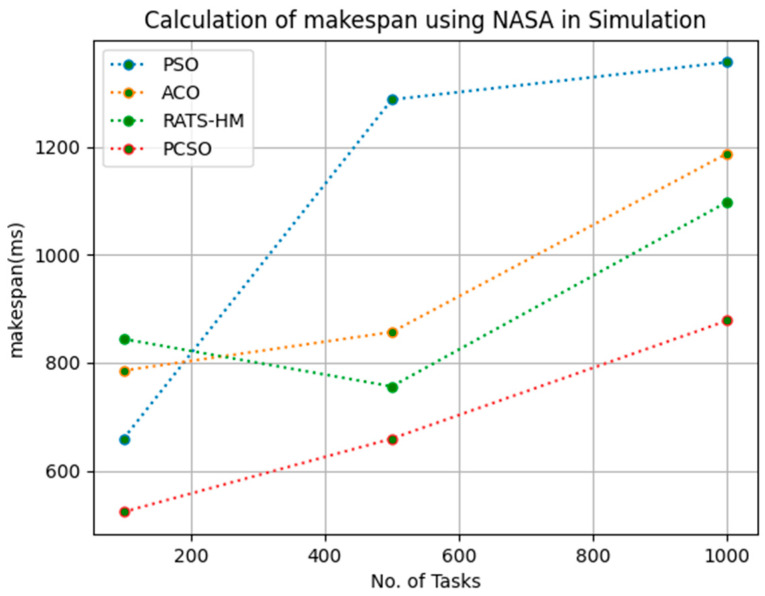
Evaluation of makespan NASA in simulation.

**Figure 6 sensors-23-06155-f006:**
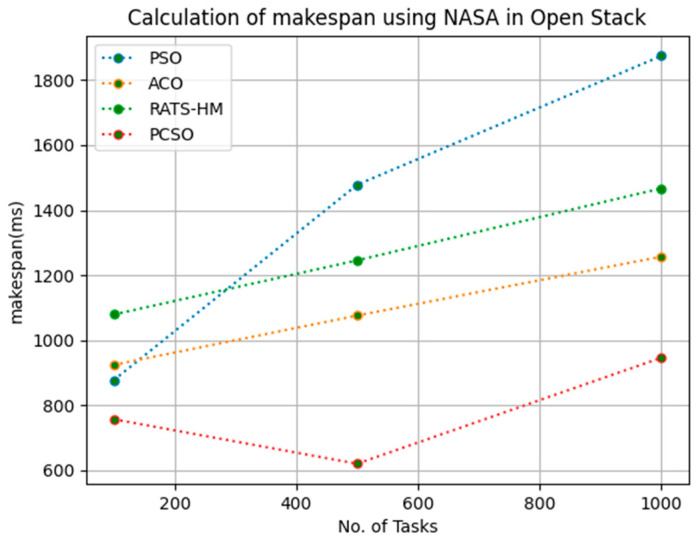
Evaluation of makespan NASA in OpenStack.

**Figure 7 sensors-23-06155-f007:**
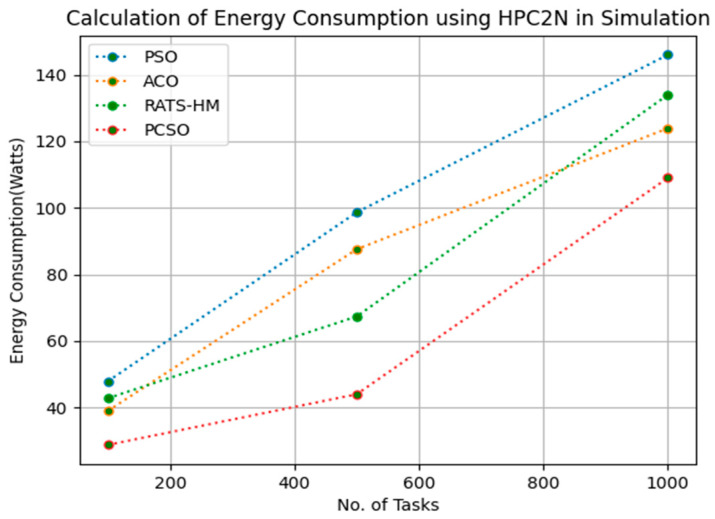
Evaluation of energy consumption HPC2N in simulation.

**Figure 8 sensors-23-06155-f008:**
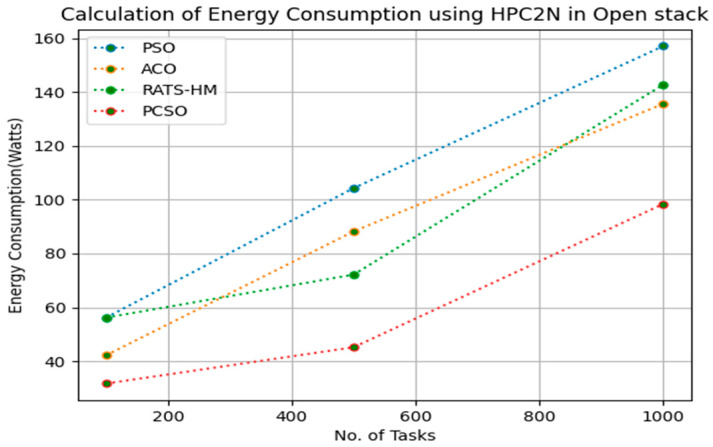
Evaluation of energy consumption HPC2N in Open stack.

**Figure 9 sensors-23-06155-f009:**
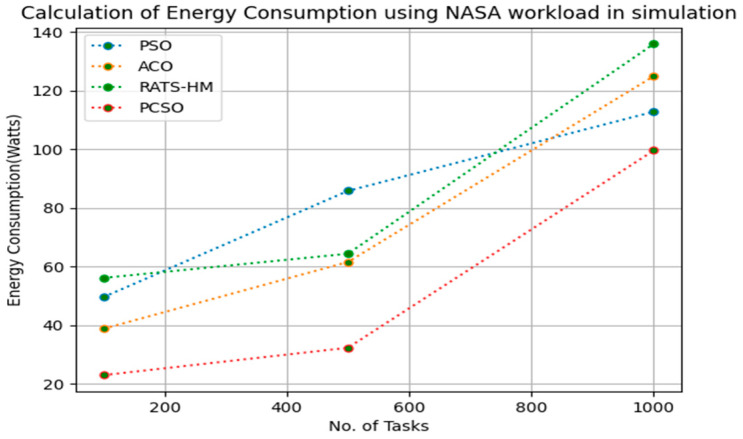
Evaluation of energy consumption NASA in simulation.

**Figure 10 sensors-23-06155-f010:**
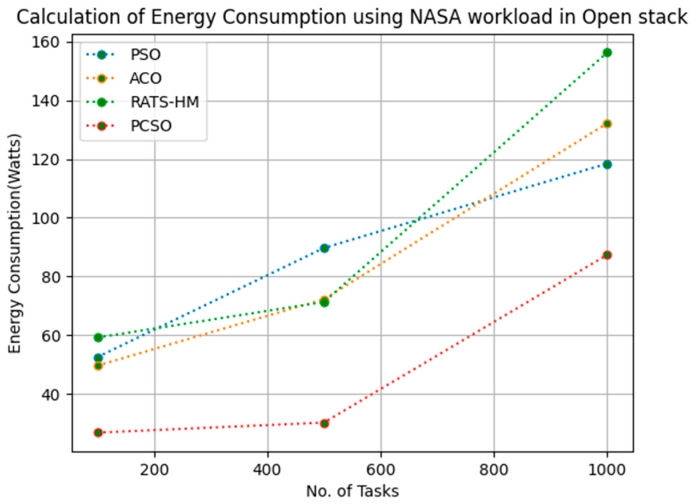
Evaluation of energy consumption NASA in OpenStack.

**Figure 11 sensors-23-06155-f011:**
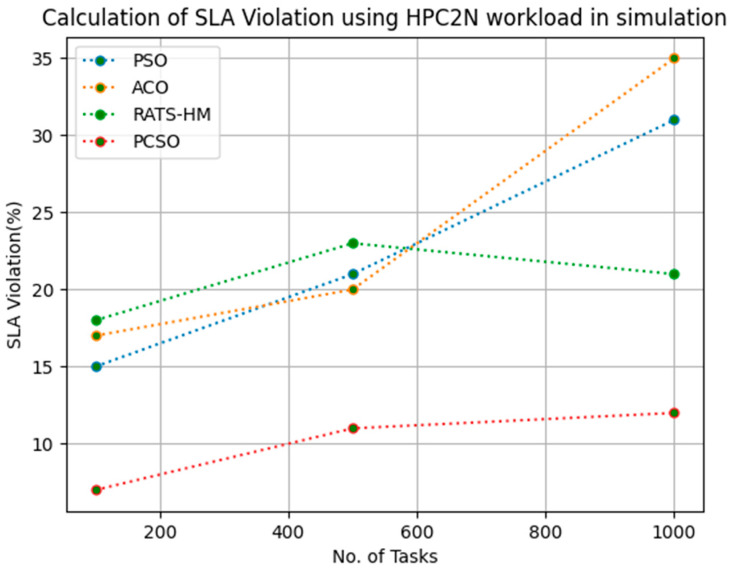
Evaluation of SLA violations HPC2N in simulation.

**Figure 12 sensors-23-06155-f012:**
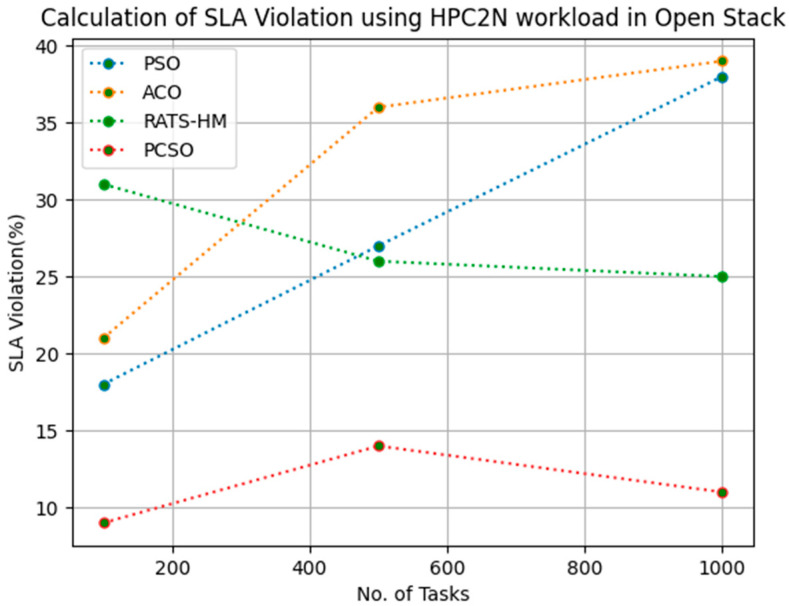
Evaluation of SLA violation HPC2N in OpenStack.

**Figure 13 sensors-23-06155-f013:**
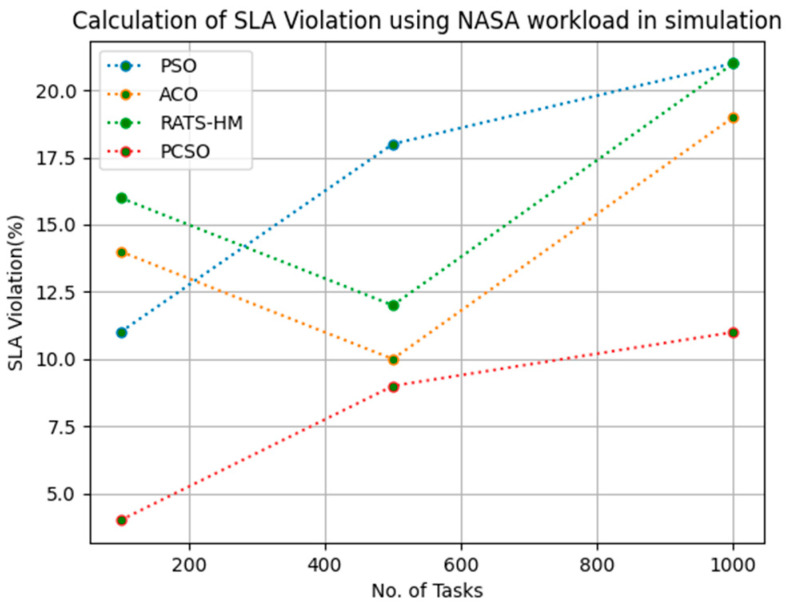
Evaluation of SLA violations NASA in simulation.

**Figure 14 sensors-23-06155-f014:**
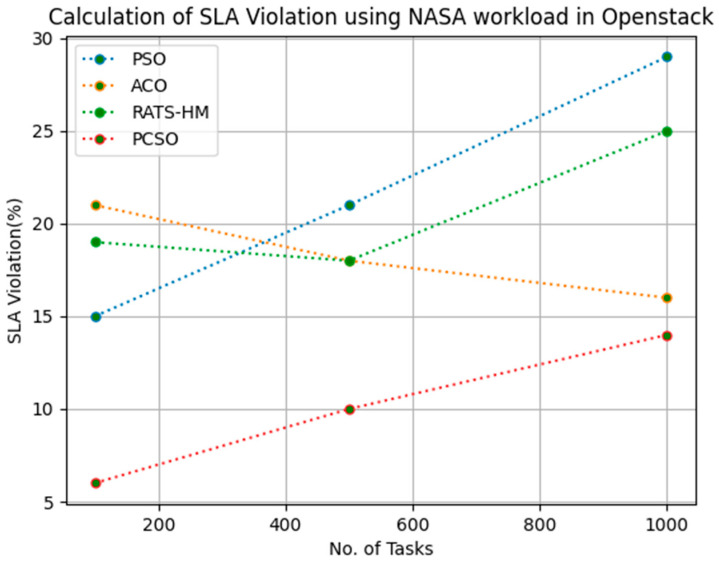
Evaluation of SLA violations NASA in OpenStack.

**Table 1 sensors-23-06155-t001:** Task-scheduling algorithms using various metaheuristic approaches.

References	Methodology	Objectives Addressed by Existing Algorithms
[6]	CSSA	Resource utilization, Energy consumption, cost, SLA violation.
[7]	CSA	Makespan.
[8]	HAPSO	Makespan, energy consumption.
[9]	MCT-PSO, LJFP-PSO	Makespan, degree of imbalance.
[10]	HIGA	Makespan, energy consumption, execution overhead.
[11]	BWF-TOPSIS	Makespan, energy consumption.
[12]	MHBFA	Makespan, energy consumption.
[13]	IACO	Energy consumption, response time, SLA violation.
[14]	EMVO	Execution time, resource utilization.
[15]	CSSA	Makespan, load balance during task distribution.
[16]	PCGWO	Makespan, cost, deadline.
[17]	MSDE	Makespan.
[18]	MVO-GA	Service availability, scalability.
[19]	ACO-Fuzzy	User costs, effective distribution of tasks to VMs.
[20]	ACO-Fuzzy	Quality of Service metrics
[24]	VMPMOPSO	SLA violation, power consumption.
[25]	SLNO	Makespan, cost, energy consumption, degree of imbalance.
[26]	VWOA	Makespan, degree of imbalance.
[27]	Crowd Sourcing platform	Cloud monitoring, real-time network monitoring.
[28]	DO4A	Job processing, transmission delay.
[29]	Adaptive resource allocation	Resource allocation, response time.
[30]	REMIX	Latency optimization in finding images in edge devices.
[31]	RATS-HM	Makespan, energy consumption, response time.
[32]	EDA-GA	Task completion time, load balancing.
[33]	Openstack	Quality of Service
[34]	HS	Utilization of resources, completion time.
[35]	HESGA	Makespan, cost, utilization of resources.
[36]	MRQFLDA	Task processing time, turnaround time.
[37]	GA-EC	Energy consumption, time delay0
[38]	HGA-ACO	Quality of service.
[39]	Adaptive load balancing	Makespan, SLA violation.
[40]	MGGS	Response time, total completion time.
[41]	EPETS	Energy consumption.
[42]	TSGA	Execution cost, resource utilization.
[43]	Agile scheduling model	Throughput.
[44]	Firefly-GA	Energy consumption.
[45]	Energy conscious GA	Energy consumption.
[46]	MFTGA	Reliability, latency, failure rate.
[47]	FHCS	Energy consumption, resource utilization.
[48]	HGSWC	Makespan.
[49]	IPSO	Makespan.
[50]	Integer PSO	Makespan, Cost

**Table 2 sensors-23-06155-t002:** Notations used in proposed System Architecture.

Notation of Entity	Meaning of Entity
$t_{n}$	No. of tasks
$v_{k}$	No. of VMs
$h_{j}$	No. of hosts
$d_{i}$	No. of Datacenters
$l^{vm}$	Load on VMs
$l^{h}$	Load on hosts
$pr^{vm}$	vm processing capacity
$t^{pr}$	Task priorities
$vm^{pr}$	Priorities of vms based on unit cost of electricity.
$ms^{n}$	Makespan of tasks
$e_{con}$	Energy consumption

**Table 3 sensors-23-06155-t003:** Settings for simulation.

Entity Name	Quantity
Tasks	1000
Task length	700,000
Ram of host	32 GB
Host storage capacity	1024 GB
Network bandwidth	500 Mbps
No. of VMs	15
RAM of VM	500 MB
Bandwidth of VM	10 Mbps
VMM used	Xen
OS	Linux
No. of datacenters	10

**Table 4 sensors-23-06155-t004:** Calculation of makespan using HPC2N in simulation.

Tasks	PSO	ACO	RATS-HM	Prioritized CSO
100	1358.9	1364.8	1486.32	1276.9
500	1756.9	1784.9	1856.18	1356.5
1000	2067.2	2245.9	2563.9	1856.8

**Table 5 sensors-23-06155-t005:** Calculation of makespan using HPC2N in OpenStack cloud.

Tasks	PSO	ACO	RATS-HM	Prioritized CSO
100	1467.7	1387.23	1567.12	1345.35
500	1768.5	1894.36	1923.98	1467.12
1000	2156.8	2256.72	2734.26	1756.21

**Table 6 sensors-23-06155-t006:** Calculation of makespan using NASA in simulation.

Tasks	PSO	ACO	RATS-HM	Prioritized CSO
100	659.2	785.56	843.98	523.67
500	1287.5	856.9	756.18	659.45
1000	1356.8	1187.92	1098.2	878.23

**Table 7 sensors-23-06155-t007:** Calculation of makespan using NASA in OpenStack cloud.

Tasks	PSO	ACO	RATS-HM	Prioritized CSO
100	876.32	923.45	1078.57	756.21
500	1478.12	1075.32	1245.32	619.17
1000	1875.11	1256.88	1467.21	945.67

**Table 8 sensors-23-06155-t008:** Calculation of energy consumption using HPC2N in simulation.

Tasks	PSO	ACO	RATS-HM	Prioritized CSO
100	47.87	38.98	42.78	28.78
500	98.65	87.56	67.36	43.99
1000	145.98	123.89	133.97	108.99

**Table 9 sensors-23-06155-t009:** Calculation of energy consumption using HPC2N in OpenStack cloud.

Tasks	PSO	ACO	RATS-HM	Prioritized CSO
100	56.15	42.15	56.18	31.67
500	104.32	88.23	72.18	45.19
1000	157.12	135.67	142.78	98.45

**Table 10 sensors-23-06155-t010:** Calculation of energy consumption using NASA in simulation.

Tasks	PSO	ACO	RATS-HM	Prioritized CSO
100	49.56	38.78	56.12	22.98
500	85.79	61.56	64.37	32.32
1000	112.79	124.89	135.88	99.56

**Table 11 sensors-23-06155-t011:** Calculation of energy consumption using NASA in OpenStack cloud.

Tasks	PSO	ACO	RATS-HM	Prioritized CSO
100	52.44	49.56	59.15	26.74
500	89.67	72.19	71.25	30.16
1000	118.43	132.18	156.28	87.34

**Table 12 sensors-23-06155-t012:** Calculation of SLA violation using HPC2N in simulation.

Tasks	PSO	ACO	RATS-HM	Prioritized CSO
100	15	17	18	7
500	21	20	23	11
1000	31	35	21	12

**Table 13 sensors-23-06155-t013:** Calculation of SLA violations using HPC2N in OpenStack cloud.

Tasks	PSO	ACO	RATS-HM	Prioritized CSO
100	18	21	31	9
500	27	36	26	14
1000	38	39	25	11

**Table 14 sensors-23-06155-t014:** Calculation of SLA violations using NASA in simulation.

Tasks	PSO	ACO	RATS-HM	Prioritized CSO
100	11	14	16	4
500	18	10	12	9
1000	21	19	21	11

**Table 15 sensors-23-06155-t015:** Calculation of SLA violations using NASA in OpenStack.

Tasks	PSO	ACO	RATS-HM	Prioritized CSO
100	15	21	19	6
500	21	18	18	10
1000	29	16	25	14

**Table 16 sensors-23-06155-t016:** Improvement of makespan over existing algorithms with various workloads in simulation.

Type of Workload	PSO	ACO	RATS-HM
HPC2N [22]	13.09%	15.92%	23.97%
NASA [23]	34.84%	27.47%	23.6%

**Table 17 sensors-23-06155-t017:** Improvement of energy consumption over existing algorithms with various workloads in simulation.

Type of Workload	PSO	ACO	RATS-HM
HPC2N [22]	40.2%	29.31%	25.55%
NASA [23]	42.55%	36.17%	39.59%

**Table 18 sensors-23-06155-t018:** Improvement of SLA violations over existing algorithms with various workloads in simulation.

Type of Workload	PSO	ACO	RATS-HM
HPC2N [22]	54.07%	56.51%	51.59%
NASA [23]	53.74%	41.16%	46.96%

**Table 19 sensors-23-06155-t019:** Improvement of makespan over existing algorithms with various workloads in OpenStack cloud.

Type of Workload	PSO	ACO	RATS-HM
HPC2N [22]	37.6%	17.5%	26.61%
NASA [23]	45.12%	28.7%	38.77%

**Table 20 sensors-23-06155-t020:** Improvement of energy consumption over existing algorithms with various workloads in OpenStack cloud.

Type of Workload	PSO	ACO	RATS-HM
HPC2N [22]	44.8%	34.11%	35.35%
NASA [23]	44.6%	43.19%	49.68%

**Table 21 sensors-23-06155-t021:** Improvement of SLA violations over existing algorithms with various workloads in OpenStack cloud.

Type of Workload	PSO	ACO	RATS-HM
HPC2N [22]	59.03%	64.59%	58.54%
NASA [23]	53.83%	45.44%	51.59%

## Data Availability

Not applicable.

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
