# Peer review of "Prioritized Task-Scheduling Algorithm in Cloud Computing Using Cat Swarm Optimization"

_sensors, 2023, doi:10.3390/s23136155_

Round 1
Reviewer 1 Report
This work presents a Cat Swarm Optimization to schedule the tasks in Cloud. The authors analyze the complexity and evaluate the performance of the proposed algorithm. Here the reviewer has some comments and hopes they are helpful in improving the manuscript quality.
1. The proposed algorithm generates the population randomly. It means the solution might converge to a different result. So, please consider using the average results, e.g. the makespan, for multiple rounds to improve the fairness.
2. In the discussion of the results, please consider more details. For example, different approaches like Eq. 15 impact the result quality.
The writing is easy to read.
Author Response
Reviewer 1:
This work presents a Cat Swarm Optimization to schedule the tasks in Cloud. The authors analyse the complexity and evaluate the performance of the proposed algorithm. Here the reviewer has some comments and hopes they are helpful in improving the manuscript quality.
- The proposed algorithm generates the population randomly. It means the solution might converge to a different result. So, please consider using the average results, e.g. the makespan, for multiple rounds to improve the fairness.
Answer: Thank you for suggestion of the reviewer. Yes, we have changed the population generation and taken average results for all metrics we have considered and we have improved results by conducting experiments in open stack cloud environment. All these are updated in manuscript and highlighted in yellow colour.
- In the discussion of the results, please consider more details. For example, different approaches like Eq. 15 impact the result quality.
Answer: Thank you for suggestion of the reviewer. We have improved results by considering a real time cloud environment and evaluated efficacy of our approach and they were updated and highlighted in manuscript in yellow colour.
Reviewer 2 Report
This paper presents a Prioritized Task Scheduling algorithm in Cloud Computing
using Cat Swarm Optimization, which are based on the all areas of cloud computing. Thus, this paper is directly related to the theme of this journal.
Overall, the paper is organized properly; the concept and future research directions are extensively explained. So, the paper is accepted after following minor changes:
1. Abstract is too large better to reduce below 200 words
2. Problem and motivation is not clear in introduction
3. First paragraph of literature review is based on the one big paragraph, its better to divide multiple paragraphs for easiness of readers
4. Add pseudo code of algorithms and Comments of the algorithm’s statement, add comments which will make more readability
5. Figure 2 are not clear, redraw figures and provide clear images
6. Paper contains few grammar mistakes which will be cooperated in final version.
7. Only 30 references are used which are very less number of references. Its better to increase references upto 50
8. add the reference related to cloud computing, which is mentioned below
Laghari, Asif Ali, Hui He, Asiya Khan, Rashid Ali Laghari, Shoulin Yin, and Jiachi Wang. "Crowdsourcing platform for QoE evaluation for cloud multimedia services." Computer Science and Information Systems 00 (2022): 38-38.
quality of english is good
Author Response
Reviewer 2:
- Abstract is too large better to reduce below 200 words.
Answer: Thank you for suggestion of the reviewer and we have updated the abstract as per your suggestion and it is highlighted in manuscript in yellow colour.
- Problem and motivation is not clear in introduction.
Answer: Thank you very much for the reviewer for suggestion and yes we have added a separate subsection as Motivation and Contributions in Introduction and it is highlighted in yellow colour.
- First paragraph of literature review is based on the one big paragraph, its better to divide multiple paragraphs for easiness of readers.
Answer: Thank you reviewer for alerting us. We have updated the paragraphs in literature as per your suggestion.
- Add pseudo code of algorithms and Comments of the algorithm’s statement, add comments which will make more readability.
Answer: Thank you reviewer for your suggestion. We have added pseudocode to algorithm and added comments in the algorithm.
- Figure 2 are not clear, redraw figures and provide clear images.
Answer: Thank you reviewer for your suggestion. We have updated the Figure 2 and not only the figure 2 we have changed all the figures and make them to be appear in better resolution.
- Paper contains few grammar mistakes which will be cooperated in final version.
Answer: Thank you reviewer for alerting us. Yes, we have proofread the entire paper once again to avoid grammatical and typos.
- Only 30 references are used which are very less number of references. Its better to increase references upto 50.
Answer: Thank you reviewer for alerting us. We have increased references in our updated manuscript and they are highlighted in yellow colour.
- add the reference related to cloud computing, which is mentioned below
Laghari, Asif Ali, Hui He, Asiya Khan, Rashid Ali Laghari, Shoulin Yin, and Jiachi Wang. "Crowdsourcing platform for QoE evaluation for cloud multimedia services." Computer Science and Information Systems 00 (2022): 38-38.
Answer: Than you reviewer for your suggestion. We have added the above reference in our updated manuscript and it is highlighted in yellow colour.
Reviewer 3 Report
Summary:
This paper presents a novel task scheduling algorithm for cloud computing that considers task priorities and utilizes the Cat Swarm Optimization (CSO) algorithm. The algorithm aims to minimize makespan, energy consumption, and SLA violation in resource provisioning. By incorporating task priorities and VM selection based on calculated priorities, the proposed algorithm effectively improves scheduling efficiency. Extensive simulations using real-time workloads compared the algorithm with baseline particle swarm optimization and ant colony optimization, demonstrating significant improvements in makespan, energy consumption, and SLA violation.
Strengths:
1. The paper is overall well-written and complete. The formulations are clear and easy to follow and the algorithm is discussed in detail. The method is clearly and adequately described and demonstrated.
2. The authors innovatively applying the Cat Swarm Optimization algorithm to the prioritized task scheduling scenario in cloud computing.
3. The authors conducted extensive experiments and compared their proposed method with existing scheduling algorithms, demonstrating its superiority in various metrics.
Weaknesses:
1. The contribution of this paper is insufficient. The Cat Swarm Optimization algorithm is a well-established algorithm, and the Prioritized Task Scheduling scenario has already been extensively researched. The authors merely applied the Cat Swarm Optimization algorithm in the context of cloud computing without significant novelty. Therefore, the significance of this work is limited.
2. The paper lacks comparison with state-of-the-art methods. In addition to comparing the proposed method with well-established methods such as PSO and ISO, it is necessary to include state-of-the-art methods as baselines.
3. The authors failed to cite several past literature (e.g., [1-3]) highly related to this work, and clearly discuss the differences between them and this paper.
[1] Online Approximation Scheme for Scheduling Heterogeneous Utility Jobs in Edge Computing, TON.
[2] MIRAS: Model-based Reinforcement Learning for Microservice Resource Allocation over Scientific Workflows, ICDCS.
[3] Flexible High-resolution Object Detection on Edge Devices with Tunable Latency, MobiCom.
4. The proposed method in this paper has only been validated in a simulated environment. It is worth considering conducting small-scale deployments in real systems.
refer to detailed comments
Author Response
Strengths:
- The paper is overall well-written and complete. The formulations are clear and easy to follow and the algorithm is discussed in detail. The method is clearly and adequately described and demonstrated.
Answer: Thank you reviewer for your compliment. Yes we clearly demonstrated the paper in a detailed manner.
- The authors innovatively applying the Cat Swarm Optimization algorithm to the prioritized task scheduling scenario in cloud computing.
Answer: Thank you reviewer for your compliment.
- The authors conducted extensive experiments and compared their proposed method with existing scheduling algorithms, demonstrating its superiority in various metrics.
Answer: Thank you reviewer for your compliment. Yes, we have conducted extensive experiments and compared their proposed method with existing scheduling algorithms
Weaknesses:
- The contribution of this paper is insufficient. The Cat Swarm Optimization algorithm is a well-established algorithm, and the Prioritized Task Scheduling scenario has already been extensively researched. The authors merely applied the Cat Swarm Optimization algorithm in the context of cloud computing without significant novelty. Therefore, the significance of this work is limited.
Answer: Thank you for suggestion of the reviewer yes many of the authors proposed various task scheduling algorithms but very few authors concentrated on consideration of priorities based on both tasks and VMs. Moreover, we have conducted both simulations and experiments in realtime cloud environment as per your suggestion in the last comment.
- The paper lacks comparison with state-of-the-art methods. In addition to comparing the proposed method with well-established methods such as PSO and ISO, it is necessary to include state-of-the-art methods as baselines.
Answer: Thank you for suggestion of the reviewer. As per your suggestion we have compared our proposed approach with state-of-art approaches by doing both simulation and experimentations in realtime cloud environment i.e. open stack.
- The authors failed to cite several past literature (e.g., [1-3]) highly related to this work, and clearly discuss the differences between them and this paper.
[1] Online Approximation Scheme for Scheduling Heterogeneous Utility Jobs in Edge Computing, TON.
[2] MIRAS: Model-based Reinforcement Learning for Microservice Resource Allocation over Scientific Workflows, ICDCS.
[3] Flexible High-resolution Object Detection on Edge Devices with Tunable Latency, MobiCom.
Answer: Thank you for suggestion of the reviewer. As per your suggestion, the above references are cited in our manuscript and they are highlighted in yellow colour in our manuscript.
- The proposed method in this paper has only been validated in a simulated environment. It is worth considering conducting small-scale deployments in real systems.
Answer: Thank you for suggestion of the reviewer. Yes, Initially we have validated our approach in simulation environment. After your comment, we have done our experiments in Open stack cloud environment which is a real time environment.
Round 2
Reviewer 1 Report
All issues have been addressed. Before publication, please authors check the description in line 303 "\\ tasks, VMs, physical hosts, data centers values initialized." Do you think this is the comment? If yes, it should be "//."
Some writing should be more carefully confirmed again.
Author Response
Reviewer 1:
- All issues have been addressed. Before publication, please authors check the description in line 303 "\\ tasks, VMs, physical hosts, data centers values initialized." Do you think this is the comment? If yes, it should be "//."
Answer: Thank you for the reviewer for your suggestion and yes we have updated the comments symbol at the line 303 and it is highlighted and updated in manuscript in yellow colour.
Reviewer 3 Report
The authors addressed my comments. I recommend acceptance.
n/a
Author Response
Reviewer 3:
- The authors addressed my comments. I recommend acceptance.
Answer: Thank you very much for the reviewer for the compliment and Yes we have addressed all the comments of the reviewer.